| 1           | Airborne polarimetric Doppler weather radar: Trade-offs between various                                        |
|-------------|----------------------------------------------------------------------------------------------------------------|
| 2           | engineering specifications                                                                                     |
| 3           | Jothiram Vivekanandan, and Eric Loew                                                                           |
| 4           | National Center for Atmospheric Research (NCAR)**                                                              |
| 5<br>6<br>7 | Boulder, CO, USA                                                                                               |
| 8           | Abstract NCAR/EOL is investigating potential configurations for the next generation airborne phased            |
| 9           | array radar (APAR) that is capable of retrieving dynamic and microphysical characteristics of clouds and       |
| 10          | precipitation. The APAR will operate at C-band. The APAR will use the electronic scanning (e-scan)             |
| 11          | feature to acquire the optimal number of independent samples for recording research quality measurements.      |
| 12          | Since the airborne radar has only a limited time for collecting measurements over a specified region           |
| 13          | (moving aircraft platform ~100 m/s), beam multiplexing will significantly enhance its ability to collect       |
| 14          | high-resolution, research quality measurements. Beam multiplexing reduces errors in radar measurements         |
| 15          | while providing rapid updates of scan volumes. Beamwidth depends on the size of the antenna aperture.          |
| 16          | Beamwidth and Directivity of elliptical, circular and rectangular antenna apertures are compared and radar     |
| 17          | sensitivity is evaluated for various polarimetric configurations and transmit/receive elements. In the case of |
| 18          | polarimetric measurements, alternate transmit with alternate receive (single channel receiver) and             |
| 19          | simultaneous reception (dual channel receiver) is compared. From an overall architecture perspective,          |
| 20          | element level digitization of transmit/receive (T/R) module versus digital sub-array is considered with        |
| 21          | regard to flexibility in adaptive beamforming, polarimetric performance, calibration, and data quality.        |
| 22          | Methodologies for calibration of the radar and removing bias in polarimetric measurements are outlined.        |
| 23          | The above-mentioned engineering options are evaluated for realizing an optimal APAR system suitable for        |

<sup>\*\*</sup> NCAR is sponsored by the National Science Foundation.

measuring the high temporal and spatial resolutions of Doppler and polarimetric measurements of 25 precipitation and clouds.

*Keywords— radar; phased array; airborne; sensing; Doppler, polarization; microphysics; weather; digital; analog; design; architecture*

1 Introduction
Characterizing location, intensity, and motion of hurricanes, tornados, and extreme precipitation events,
and understanding effects of clouds and aerosols on the earth radiation budget requires a better
understanding of the kinematic (storm motion and structure) and microphysical processes (particle growth,
phase changes) within these storms. This remains a challenge for both the scientific and operational
communities. Observing these events is challenging using ground-based radars as the conditions that lead
to high-impact events typically occur in remote areas (e.g., hurricanes over the ocean, orographic
precipitation in rugged terrain) or because of large uncertainties on the timing and location (e.g., tornado,
extreme precipitation events) related to sub-optimal radar coverage. Airborne radar is a powerful tool to
observe weather systems, in particular, storms over complex terrain, the ocean, polar regions, and forest
regions not easily observable by ground-based radars (Bluestein and Wakimoto, 2003). A scanning Doppler
radar on an airborne platform is used for estimating dual-Doppler winds with the help of rapid scanning as
the aircraft flies past a storm (Hildebrand et al. 1996). Scanning Doppler radar with dual-polarization
capability on an airborne platform is capable of measuring dual-Doppler winds and retrieving particle types
(ice or water) and shapes, and liquid/ice water contents using reflectivity (Z), differential reflectivity (Z<sub>DR</sub>),
specific propagation phase (K<sub>DP</sub>), and linear depolarization ratio (LDR).

At present, no other instrument other than an airborne polarimetric Doppler phased array radar system has
the potential to estimate high temporal and spatial measurements of 3-D winds and microphysics
concurrently (Vivekanandan et al. 2014). NCAR's Earth Observing Laboratory (EOL) is currently
conducting a design study for a future airborne phased array radar.
Between 1992 and 2013 NCAR operated research quality Doppler radar, ELDORA/ASTRAIA (Electra 51 Doppler Radar/Analysee Steroscopic par Impulsions Aeroporte, hereafter referred as ELDORA). The 52 ELDORA was configured with dual slotted waveguide array antennas using dual-transmitter, dual-beam, 53 rapid scan and step-chirped waveform (Girardin-Gondeau et al. 1991) that significantly improved the along-54 track spatial resolution from 750 m to 300 m when compared to NOAA's airborne tail Doppler radar (TDR) 55 (Hildebrand et al. 1996). The ELDORA was jointly developed by NCAR and the Center de Recherché en 56 Physique de l'Environment Terrestre et Planetaire, France. It collected research quality Doppler and 57 reflectivity measurements that continue to set the standard for airborne radar; however, ELDORA X-band 58 radar's penetration into precipitation is limited by attenuation and it is not designed to collect polarimetric 59 measurements to remotely estimate microphysics. ELDORA has been placed in dormancy because its 60 airborne platform (Naval Research Lab P-3 587) was retired in January 2013. The US research community 61 has strongly voiced the need to continue measurement capability similar to that provided by ELDORA 62 (Smith et al. 2012).





The combination of remote and *in situ* sensors on a single airborne platform will serve the observational 72 needs for broader scientific communities of cloud microphysics, mesoscale meteorology, atmospheric 73 chemistry and climate, and it will fill a critical gap in the current airborne observing facilities. APAR 74 deployed on a long on-station time aircraft, such as the C-130, will allow investigation of weather systems 75 such as monsoons, tropical cyclones, severe convection over continents, orographic precipitation, 76 convection over the oceans, and polar and low to middle atmospheric chemistry. A schematic of the APAR 77 antenna panels on the C-130 is shown in Figure 1. APAR will feature four removable C-band active 78 electronically scanned arrays (AESA) mounted on top, both sides, and the bottom of the aircraft. Each 79 antenna will have dual-Doppler and polarimetric capabilities. This configuration, when integrated with data

Figure 1. Notional schematic of APAR AESA antenna panel placement on the C130. There are two sidepanels on port and starboard of the fuselage aft of the rear personnel doors.

from the NSF/NCAR C-130 nose radar will provide 360-degree horizontal surveillance radar scan coverage and volumetric data collection. Radar products can be displayed in real time. The AESA placement provides for flexible scanning capabilities for 3-D data volume generation.

83 The real-time radar displays on the aircraft provide situational awareness to aircraft pilots allowing for 84 safe aircraft operations in the vicinity of extreme weather. The 3-D volume-scan data not only can help 85 guide the NSF/NCAR C-130 research in and around weather of interest, but also has the potential to be used 86 to guide other aircraft conducting research in the vicinity. The C130 is a versatile and capable research 87 platform that carries a wide variety of scientific payloads. The C130 has a 10-hour flight endurance, a 2,900 88 nautical mile range at up to 27,000 ft., and a payload capacity of up to 13,000 lbs. NCAR EOL/RAF 89 maintains the NSF/NCAR C130 aircraft in its fleet for airborne atmospheric measurements, including 90 dropsonde, in situ sampling and remote sensing of clouds, chemistry, and aerosols.

91 This paper is organized as follows. Section 2 describes radar system, and its major sub-systems. Rationale 92 for selection of transmit frequency is presented in Section 3. Discussions related to polarimetric 93 configurations and advantage of agile beam scanning is presented in Sections 4 and 5. Various antenna 94 aperture configurations and corresponding beamwidth characteristics are presented in Section 6. The 95 sensitivity of the radar measurements depends on transmit and receive hardware characteristics, polarimetric 96 measurement configuration, and signal processing. Expected radar sensitivity as a function of a few key 97 parameters is discussed in Section 7. PAR with digital architecture is amenable for consistently maintaining 98 data quality, deployment of radar with repeatable and robust calibration and also formation of fan-beam and 99 pencil beam configuration for imaging rapidly changing weather system. In this regard possible analog and 100 digital beamforming architecture configurations are compared in Section 8. A brief description of 101 calibration requirement is discussed in Section 9. Estimates of biases in radar measurements as a result of 102 cross coupling in polarimetric mode is shown in Section 10. Section 11 presents a summary.

## 104 **2. System Description**

105

106 Preliminary design specifications of peak power, beamwidth, dual-polarimetric configuration, scan timing 107 sequences and signal processing are outlined in Vivekanandan et al. 2014; some key technical specifications 108 are presented in Table 1. The APAR will operate at C-band. It will use the e-scan feature to acquire the 109 optimal number of independent samples for achieving 1 m/s accuracy in radial velocity, 1 dB in reflectivity 110 and 0.2 dB in differential reflectivity accuracies with a sensitivity of -11 dBZ at 10 km. Since the airborne 111 radar has only a limited time for collecting measurements over a specified region (moving aircraft platform 112  $\sim 100$  m/s), beam multiplexing will significantly enhance its ability to collect high-resolution, research 113 quality measurements. Beam multiplexing reduces errors in radar measurements while providing rapid 114 updates of scan volumes (Weber et al. 2007, Yu et al. 2007).

115 From an overall architecture perspective, element level digitization of T/R module versus digital sub-116 array has to be carefully considered with regard to flexibility in adaptive beamforming, polarimetric 117 performance, calibration, and imaging of rapidly moving weather system. For achieving desired sensitivity 118 and range resolution, pulse compression is proposed, with a compression gain of at least 15 dB. However, 119 this will require transmission of short pulses for covering the blind zone created by the strong pulse at the 120 expense of overall radar sensitivity due to long and short pulses. A staggered pulse repetition frequency 121 (PRF) technique for extending Doppler Nyquist interval was extremely valuable for the ELDORA is 122 considered, but its evaluation is beyond the scope of this document.

123

# 124**Table 1.** Technical Specifications of C-band APAR

| Parameter           | Numeric value                                |
|---------------------|----------------------------------------------|
| Operating Frequency | C-band: 5.35 - 5.45 GHz (FAA<br>requirement) |