# Peer review of "measuring the high temporal and spatial resolutions of Doppler and polarimetric measurements of 25 precipitation and clouds."

_Geoscientific Instrumentation, Methods and Data Systems, 2017_

## Referee Comment (RC1) · J. TESTUD (Referee) · 16 Oct 2017

This paper presents a very ambitious project of airborne polarimetric Doppler radar, as a follow on of the previous ELDORA/ASTRAIA radar developed in the 90's between NCAR and CNRS. I approve the main definition features of the project: - C-130 as the aircraft carrier, - Phased array antenna for the radar technology, - Polarisation diversity capability. Very good paper. I have nevertheless some questions or comments:

Table 1- and Figure 6: I understand that the choice of the radar frequency, C band (instead of X band for ELDORA/ASTRAIA) is dictated by the concern of avoiding situation of total extinction of the radar signal in severe weather. However, I am wondering if

[Figure]

this choice is not too much penalizing in antenna performance -angular resolution and side lobes. - Table 1: I do not understand how you can achieve a 3 dB beam resolution (one-way) of 2.2° with a 38" diameter antenna at C band. A reflector antenna with good sidelobes (<-30 dB), respects a relation like : 3dB_beam_res ≈ 65 $\lambda$/D (1). - With this relation, your 3 dB beam aperture should be 3.7°. Can you improve the performance predicted by rel. (1) simply because you may control much more easily the antenna illumination with phase array technique? If yes, it's worth mentioning. - Fig.6, the first sidelobe is at -15 to – 17 dB, which may be quite penalizing from airborne where part of the exploration is made at negative elevation where you must address the problem of the surface clutter. It's the reason why -30dB side lobes were specified for the EL-DORA/ASTRAIA antenna. Did you check (by simulation?) that your antenna sidelobes are compatible with your objective of detecting -10dBZ within 400m of surface at 5 km range?

Section 4: Polarimetric measurement configuration Your discussion about ATSR (alternate transmit and simultaneous receive) and STSR (Simultaneous transmit and simultaneous receive) is interesting. Today in operational, most radars use the STSR mode, since the ATSR mode requires a high-power polarization switch, a component very expensive and unreliable. A big potential interest of the phased array is that it opens the possibility of using very naturally the ATSR mode, which authorized the possibility to measure LDR (impossible with STSR). However, I totally disagree with the argument of the author to discard STSR on the argument that with this mode the "isolation" between H and V should be 44dB. How this "isolation" is defined? Is it the usual crosspolar level? In that case, that would mean the impossibility of STSR methodology since no antenna holds this performance. Meanwhile hundreds of operational polarimetric radars provide satisfactory data (including ZDR) worldwide. In fact, the criteria for appropriate measurement of ZDR with STSR is the same as the one for LDR. It is based of the same ICPR cited by the authors in their formula (1). Simply, It is less stringent with STSR to measure ZDR than with ATSR to measure LDR. I figured out that to measure LDR down to – 27 dB with ATSR, ICPR should be below -33 dB (as

recommended by Bringi and Chandraseckar, 2001), while to measure ZDR with 0,2 dB bias with STSR, ICPR should be below -23 dB (In the extreme case where ZDR$\approx$ -10dB (due to differential attenuation). I think it would be wise within this project to maintain the capability of the system to operate polarimetric measurements both with ATSR and STSR methodologies.

Jacques Testud, October 16 th, 2017

---

## Referee Comment (RC2) · Anonymous Referee #2 · 17 Oct 2017

**Jothiram Vivekanandan and Eric Loew**

**Anonymous Referee #2**

The paper presents a potential configuration of the next Generation of an APAR. A lot of nice technical details were written, the paper is nice to read and understood. I recommend to publish this paper after some modifications: 1- First of all, I think this paper should be in a series of 3 papers, the first is this one, the second will be to verify all the parametrization for a test period and studying the stability, and the third one for the results which will be more scientific than technic. Here are other remarks: L9: microphysical characteristics of clouds at C-band!! It is too ambitious... Maybe dual-pol characteristics for precipitations but not for clouds!! L55 – L56: I believe you mean CETP: Centre d'étude des Environments Terrestre et planétaires. By the way, this lab was mixed with another laboratory and the current name is LATMOS (Laboratoire

[Figure]

Atmospheres, Milieux, Observations Spatiales). L87-L90 (and in the entire paper): Units, It would be better to use standard units (International systems), but if authors prefer to use the American units (inch, pounds, lb, ...) it would be better to put also the classic units (cm, m, Kg, …) between brackets maybe! L109-L110 (and in the entire paper): abbreviations, it is recommendable to user abbreviations one the authors define them. L117 – L122 (Major remark): Sensitivity loss (in fig 8 too) is not been considered for Pulse compression mode and short pulse mode! The transition area (in range) must be carefully filtered and weighted! L132: Antenna is one of the most important parts and it is not really detailed. L126 (in the table): AZ and EL scan range should be shown. L189-L190: Agree, but the authors are talking about dual-pol radar, so the attenuation can be considered! Section 3: I am not really convinced by the C-band. Anyway, it is a choice but: figure 3 is not the good way to show the attenuation. It is known that S is better than C and X, same for C and X but why comparing until 100km or more, for X-band we are talking about 30 to 50 km and is the intention is to go more than 100km? For what? And on which direction? Additionally, a vertical cut through clouds by APAR shows usually less attenuation compared to a classic PPI scan from stationary radar! By the way, I believe X-band is better for microphysics! L230-L234: it is better to add a reference. L246-L248: can the authors add numerical values? L253-256: different font type. L268 and L300: 20° and 0°(typo error). L439: -100 dBm? For the polarization, authors should not only consider the polarization change of the co-polar pattern during beam steering-they should also consider the fact that the pol pattern rises when the beam is steered. To avoid that differential feeding should be applied for the phase excitation of the array must be used. I invite the authors to read these 2 references for more details: - Vollbracht, D.: Understanding and optimizing microstrip patch antenna cross polarization radiation of element level for demanding phased array antennas in weather radar applications, Adv. Radio Sci., 13, 251-268, doi:10.5194/ars-13-251-2015, 2015.

- Vollbracht D.: "Optimum phase excitations and probe- feed positions inside antenna arrays for the reduction of Cross Polarization radiation in demanding phased array

weather radar applications", 10th European Conference on Antennas and Propagation (EuCAP) Switzerland, Davos, 2016

---

## Author Comment (AC1) · 2 Nov 2017

Review of GI-2017-45: Airborne polarimetric Doppler weather radar: Trade-offs between various engineering specifications Authors: Jothiram Vivekanandan, and Eric Loew

Summary: This paper presents a very ambitious project of airborne polarimetric Doppler radar, as a follow on of the previous ELDORA/ASTRAIA radar developed in the 90's between NCAR and CNRS. I approve the main definition features of the project: - C-130 as the aircraft carrier, - Phased array antenna for the radar technology, - Polarisation diversity capability. Very good paper. I have nevertheless some questions or

comments.

Reply to Reviewer #1

Dear Reviewer,

Thank you for your time and comments toward helping us improve the manuscript. We have revised the manuscript based on your comments. The revisions made to the manuscript are as follows.

Table 1- and Figure 6: I understand that the choice of the radar frequency, C band (instead of X band for ELDORA/ASTRAIA) is dictated by the concern of avoiding situation of total extinction of the radar signal in severe weather. However, I am wondering if this choice is not too much penalizing in antenna performance -angular resolution and side lobes.

A: The C-band is chosen for achieving similar angular resolution as in the ELDORA and also for keeping the cost of the radar system lower. Since in the airborne research community, the ELDORA's measurement is considered to be the standard, the desired goal for the proposed APAR is to meet the current sensitivity of the ELDORA. On the C-130 maximum, allowable antenna aperture size is 1.93 m (76"). It will produce a narrower beamwidth at X-band, but it would require four times the number of T/R elements and consequently would be more expensive.

Table 1: I do not understand how you can achieve a 3 dB beam resolution (one-way) of 2.2âŮę with a 38" diameter antenna at C band. A reflector antenna with good side-lobes (<-30 dB), respects a relation like : 3dB_beam_res $\approx$ 65 $\lambda$/D (1). - With this relation, your 3 dB beam aperture should be 3.7âŮę. Can you improve the performance predicted by rel. (1) simply because you may control much more easily the antenna illumination with phase array technique? If yes, it's worth mentioning.

A: Sorry for the confusion about antenna size. On the C-130 maximum, the allowable aperture is an ellipse of 1.93 m (76") major diameter and 1.78 m (70") minor diameter.

The specification of antenna size in Table 1 has been changed to diameter.

Fig.6, the first sidelobe is at -15 to – 17 dB, which may be quite penalizing from airborne where part of the exploration is made at negative elevation where you must address the problem of the surface clutter. It's the reason why -30dB side lobes were specified for the ELDORA/ASTRAIA antenna.

Did you check (by simulation?) that your antenna sidelobes are compatible with your objective of detecting -10dBZ within 400m of surface at 5 km range?

A: Fig. 6 shows a uniformly illuminated pattern, where no amplitude tapering is done. This yields the most antenna gain, narrowest beamwidth, and highest sidelobe levels. The intent of this figure is to compare the relative characteristics (sidelobe and mainlobe) of the antenna apertures being considered for APAR, namely elliptical, circular and square. Unlike traditional antennas, active electronically scanned antennas (AESAs), have inherent flexibility to alter the antenna pattern by adjusting both phase and amplitude at each antenna element. Typically, the antenna patterns are not symmetrical on transmit and receive. Fig. 7. illustrates the two-way, combined antenna patterns the solid red line shows that -50 dB two-way peak sidelobes are achievable using a combination of amplitude weighting on transmit (15 dB Taylor) and on receive (-30 dB Taylor). Randomizing the active elements on transmit while applying amplitude weighting on receive has been shown to further reduce the first sidelobe level to <- 55 dB. Another approach is to effectively null the near in sidelobes which intersect the ground. Both these approaches are still topics of ongoing research. Based on recent simulations, the circular aperture having -50 dB two-way peak sidelobes does not meet the objective of detecting -10 dBZ within 400m of a surface at 5 km range. In fact, this objective pushes the current capabilities of most conventional antennas and very definitely AESA antennas.

Section 4: Polarimetric measurement configuration Your discussion about ATSR (alternate transmit and simultaneous receive) and STSR (Simultaneous transmit and simultaneous receive) is interesting. Today in operational, most radars use the STSR mode, since the ATSR mode requires a high-power polarization switch, a component very expensive and unreliable. A big potential interest of the phased array is that it opens the possibility of using very naturally the ATSR mode, which authorized the possibility to measure LDR (impossible with STSR). However, I totally disagree with the argument of the author to discard STSR on the argument that with this mode the "isolation" between H and V should be 44dB. How this "isolation" is defined? Is it the usual crosspolar level? In that case, that would mean the impossibility of STSR methodology since no antenna holds this performance. Meanwhile hundreds of operational polarimetric radars provide satisfactory data (including ZDR) worldwide. In fact, the criteria for appropriate measurement of ZDR with STSR is the same as the one for LDR. It is based of the same ICPR cited by the authors in their formula (1). Simply, It is less stringent with STSR to measure ZDR than with ATSR to measure LDR. I figured out that to measure LDR down to – 27 dB with ATSR, ICPR should be below -33 dB (as recommended by Bringi and Chandraseckar, 2001), while to measure ZDR with 0,2 dB bias with STSR, ICPR should be below -23 dB (In the extreme case where ZDR≈ -10dB (due to differential attenuation). I think it would be wise within this project to maintain the capability of the system to operate polarimetric measurements both with ATSR and STSR methodologies. Jacques Testud, October 16 th, 2017

A: We agree with the reviewer's comments. The requirement of -44 dB isolation for the STSR mode is based on the worst-case scenario where the differential propagation phase varies up to 1500. The following sentences about the cross polarization isolation requirement have been added to section 4:

"Cross-polarization isolation requirement is less stringent for estimating unbiased Z and ZDR in the ATSR mode than in the STSR mode. Cross polarization isolation depends on ICPR of the radiating elements, cross-polar system phase (phase difference between co and cross-channel) and the differential propagation phase of the precipitation medium in the STSR mode. Assuming the system phase characteristic is known, ICPR

< -23 dB is required for estimating ZDR with less than 0.2 dB bias (Wang and Chandrasekar 2006). In ATSR mode ICPR < -20 dB is satisfactory for estimating ZDR with less than 0.2 dB bias. However, for ICPR better than -33 dB is required for measuring intrinsic LDR of -27 dB (Bringi and Chandraseckar, 2001)."

References

Bringi, V. N., and Chandrasekar, 2001: Polarimetric Doppler weather radar. Cambridge University press.

Wang, Y., and V. Chandrasekar, 2006: Polarization isolation requirements for linear dual-polarization weather radar in simultaneous transmission mode of operation. IEEE Trans. Geosci. Remote Sensing, 44, 2019-2028.

---

## Author Comment (AC2) · 2 Nov 2017

Review of GI-2017-45: Airborne polarimetric Doppler weather radar: Trade-offs between various engineering specifications Authors: Jothiram Vivekanandan, and Eric Loew

Summary:

The paper presents a potential configuration of the next Generation of an APAR. A lot of nice technical details were written, the paper is nice to read and understood. I recommend to publish this paper after some modifications:

[Figure]

Reply to Reviewer # 2

Dear Reviewer,

Thank you for your time and comments toward helping us improve the manuscript. We have revised the manuscript based on your comments. Following is the summary of the revision.

General comment: First of all, I think this paper should be in a series of 3 papers, the first is this one, the second will be to verify all the parametrization for a test period and studying the stability, and the third one for the results which will be more scientific than technic.

A: Thank you for your suggestion for extending the results presented in this manuscript. The authors are in the process of evaluating engineering trade-offs presented in this manuscript using an observation simulation experiment evaluation (OSEE). This study will optimize engineering specifications by simulating the APAR observations for a range of cloud model results. This effort will lead to subsequent manuscripts.

Here are other remarks:

L9: microphysical characteristics of clouds at C-band!! It is too ambitious... Maybe dualpol characteristics for precipitations but not for clouds!!

A: We agree with reviewer's comment that C-band measurements are not as good as cloud radar observations at Ka and W-bands. Nevertheless, the sensitivity of the APAR is expected to be better than -15 dBZ at ranges less than 10 km. A threshold of -17 dBZ is used to distinguish regions of cloud and drizzle (Fox and Illingworth, 1997). Remote measurements alone will not be able to distinguish regions of cloud ice and cloud liquid as the sensitivity of cross polarization observation is also a limiting factor.

L55 – L56: I believe you mean CETP: Centre d'étude des Environments Terrestre et planétaires. By the way, this lab was mixed with another laboratory, and the current name is LATMOS (LaboratoireAtmospheres, Milieux, Observations Spatiales).

A: The sentence is revised as follows: "The ELDORA was jointly developed by NCAR and the Centre d'étude des Environments Terrestre et Planétaires (CETP), France. In the recent years, CETP was merged with LATMOS (Laboratoire Atmospheres, Milieux, Observations Spatiales), France."

L87-L90 (and in the entire paper): Units, It would be better to use standard units (International systems), but if authors prefer to use the American units (inch, pounds, lb, ...) it would be better to put also the classic units (cm, m, Kg, . . .) between brackets maybe!

A: Units have been changed to the standard international system.

L109-L110 (and in the entire paper): abbreviations, it is recommendable to user abbreviations one the authors define them.

A: Thanks for the suggestion. Abbreviations are used wherever they improved the clarity of the presentation.

L117 – L122 (Major remark): Sensitivity loss (in fig 8 too) is not been considered for Pulse compression mode and short pulse mode! The transition area (in range) must be carefully filtered and weighted!

A: The text will be modified starting at line 362: "Figure 8b shows the sensitivity of the APAR as a function of range as depicted by the green curve. The discontinuity in the curve shows the sensitivity difference caused by the transition from short pulse mode to pulse compression mode. In this illustration, the short pulse region extends to 5 km in range and would be ∼9 dB less sensitive than the pulse compression region. This sensitivity loss can be mitigated by optimally positioning the aircraft to maximize the sensitivity of the areas of greatest interest. Care must also be taken to merge the data between the two regions, as pulse compression filtering effects can cause artifacts in the data if not handled properly."

L132: Antenna is one of the most important parts and it is not really detailed.

A: A brief description of the antenna has been added section 2: "Each radiating element will use a stacked patched microstrip antenna radiator coupled to transmit/receive (T/R) module. The microstrip patch antenna elements can transmit in either horizontal (H) or vertical (V) polarizations. The radiating elements are spaced less than half a wavelength apart to avoid grating lobes over the full scan extent (Wang et al. 2008)." The dual-polarized patch antenna element proposed overcomes the problems of isolation in the diagonal plane and mismatch between the horizontal and vertical co-polarizations by combining the features of a parasitic crosspatch antenna and a ground plane with a cross-shaped aperture and capacitive and inductive loading corners. The patch is fed symmetrically in both horizontal and vertical polarizations."

L126 (in the table): AZ and EL scan range should be shown.

A: Table 1 has been revised to include Az and El scan ranges.

L189-L190: Agree, but the authors are talking about dual-pol radar, so the attenuation can be considered!

A: The following sentences have been added: "In the case of rain, the specific propagation phase (KDP) is proportional to rain intensity. Attenuation (AH) and specific differential attenuation (ADP) are almost linearly proportional to KDP (Bringi et al,. 1990). As KDP is unaffected by attenuation, radar system bias due to change in transmit power, and antenna and receiver gain factors, it is more commonly used for attenuation correction."

Section 3: I am not really convinced by the Cband. Anyway, it is a choice but: figure 3 is not the good way to show the attenuation. It is known that S is better than C and X, same for C and X but why comparing until 100km or more, for X-band we are talking about 30 to 50 km and is the intention is to go more than 100km? For what? And on which direction? Additionally, a vertical cut through clouds by APAR shows usually less attenuation compared to a classic PPI scan from stationary radar! By the way, I believe X-band is better for microphysics!

A: Figure 3 is used only for a relative comparison of attenuation effects at S, C, and X-bands and show subtle differences due to Mie scattering at higher frequencies. The C-band is chosen for achieving similar angular resolution as in the ELDORA and also for keeping the cost of the radar system lower. Since in the airborne research community, the ELDORA's measurement is considered to be the standard, the desired goal for the proposed APAR is to meet the current sensitivity of the ELDORA. On the C-130 maximum, allowable antenna aperture size is 1.93 m (76"). It will produce a narrower beamwidth at X-band, but it would require four times the number of T/R elements and consequently would be more expensive.

L230-L234: it is better to add a reference.

A: Bringi and Chandrasekar, 2001 reference has been included in support of cross-polarization isolation requirement.

L246-L248: can the authors add numerical values?

A: The sentence has been revised as follows: "Second, isolation between the co-polar and cross-polar channels must be at least 6 dB greater than the desired LDR lower limit."

L253- 256: different font type.

A: The font has been corrected.

L268 and L300: 20âŮȩ and 0âŮȩ (typo error).

A: The typographical error has been corrected.

L439: -100 dBm?

A: Yes, the –ve sign has been added.

For the polarization, authors should not only consider the polarization change of the co-polar pattern during beam steering-they should also consider the fact that the pol

pattern rises when the beam is steered. To avoid that differential feeding should be applied for the phase excitation of the array must be used. I invite the authors to read these 2 references for more details: - Vollbracht, D.: Understanding and optimizing microstrip patch antenna cross polarization radiation of element level for demanding phased array antennas in weather radar applications, Adv. Radio Sci., 13, 251-268, doi:10.5194/ars-13-251-2015, 2015. - Vollbracht D.: "Optimum phase excitations and probe- feed positions inside antenna arrays for the reduction of Cross Polarization radiation in demanding phased array weather radar applications", 10th European Conference on Antennas and Propagation (EuCAP) Switzerland, Davos, 2016

A: Thank you for providing latest references for microstrip patch antenna design and development. They are helpful. The current version of the microstrip patch antenna is excited symmetrically in both horizontal and vertical polarizations. Also, to limit the effect of differential gain and beam pattern on polarimetric measurements, co-polarization measurements will be collected only up to 200 from broadside.

References

Bringi, V., V. Chandrasekar, N. Balakrishnan, and D. Zrnic, 1990: An Examination of Propagation Effects in Rainfall on Radar Measurements at Microwave Frequencies. J. Atmos. Oceanic Technol., 7, 829–840.

Bringi, V. N., and Chandrasekar, 2001: Polarimetric Doppler weather radar. Cambridge University Press.

Fox, N. I., and A. J. Illingworth, 1997: The retrieval of stratocumulus properties by ground-based radar. J. Appl. Meteor., 36, 485–492.

Wang, H., D. Fang, Y. L. Chow, 2008: Grating lobe reduction in a phased array of limited scanning. IEEE Trans. Ante. Prop., 1581-1585.